# On deeper human dimensions in Earth system analysis and modelling

Dieter Gerten[1,2], Martin Schönfeld[3], Bernhard Schauberger[1,4]

[1] Potsdam Institute for Climate Impact Research, Telegrafenberg, 14473 Potsdam, Germany
5  [2] Geography Department, Humboldt-Universität zu Berlin, 10099 Berlin, Germany
[3] Philosophy Department, University of South Florida, 4202 E. Fowler Ave., FAO 248, Tampa, FL 33620, USA
[4] Laboratoire des Sciences du Climat et de l'Environnement, Orme des Merisiers, 91191 Gif-sur-Yvette, France

*Correspondence to*: Dieter Gerten (gerten@pik-potsdam.de)

**Abstract**

While humanity is altering planet Earth at unprecedented magnitude and speed, representation of the cultural driving factors and their dynamics in models of the Earth system is limited. In this review and perspectives paper, we argue that more or less distinct environmental value sets can be assigned to religion – a deeply embedded feature of human cultures, here defined as collectively shared belief in something sacred. This assertion renders religious theories, practices and actors suitable for studying cultural facets of anthropogenic Earth system change, especially regarding deeper, non-materialistic motivations that ask about humans' self-understanding in the Anthropocene epoch. We sketch a modelling landscape and outline some research primers, encompassing the following elements: i) extensions of existing Earth system models by quantitative relationships between religious practices and biophysical processes, building on databases that allow for (mathematical) formalisation of such knowledge, ii) design of new model types that specifically represent religious morals, actors and activities as part of coevolutionary human-environment dynamics, and iii) identification of research questions of humanitarian relevance that are underrepresented in purely economic-technocratic modelling and scenario paradigms. While this analysis is by necessity heuristic and semi-cohesive, we hope that it will act as a stimulus for further, interdisciplinary and systematic research on the immaterial dimension of humanity's imprint on the Earth system, both qualitatively and quantitatively.

## 1 Introduction and Motivation

In the current Anthropocene, human interventions are transforming Earth to a degree unprecedented in civilizational history, producing manifold detrimental impacts upon the climate system, the biosphere, and the global water system, to name but a few (Steffen et al., 2015). Earth system models are designed to quantify the joint dynamics of biogeophysical, technological and also social processes at a global scale, but doubt has recently been increasing whether humans, their activities and especially their cultures are adequately considered in such models. In his seminal paper on the topic, Schellnhuber (1999) compared the mere "physiological–metabolic contribution of global civilisation to planetary operation" to the role played by, for example, the sheep inhabiting this planet with their collective impact upon the biotic and abiotic environment (by reflecting sunlight, grazing, and emitting methane). Indeed, the predominant way that humans are represented in current models of the Earth system – including Integrated Assessment Models and stand-alone global climate, ecological or hydrological models – is by way of their biogeophysical manifestation (emissions, resource use, etc.), simple metrics of societal impacts of environmental change (such as the number of people affected by water scarcity, crop yield losses, flood damages, or sea level rise), and/or equilibrium representations of the world economic system.

As Donges et al. (2017a) point out, most global models, thus, do not adequately capture the pronounced and diverse dynamics of the human component of the Earth system, hampering simulation of the coevolution of human societies and their environment with its characteristic nonlinearities (tipping points, regime shifts, emergent properties) and global teleconnections (material and information networks, cross-scale cascades, etc.). Despite substantial progress especially

regarding reduced-complexity social-ecological models and agent-based and cellular social-ecological models (Hornborg and Crumley, 2006; Dearing et al., 2012; van Vuuren et al., 2012; Caldas et al., 2015; Verburg et al., 2016; Müller-Hansen et al., 2017), human agencies, networks and complex coevolutionary dynamics are largely neglected. This underrepresentation takes the form of models reducing human behaviour to its physical manifestations, devoid of cultural drivers and dynamics.

This bears the risk of overlooking decisive socio-cultural developments, for example towards implementing the Paris Agreement, abiding by the Convention on Biological Diversity, water, energy and food security goals, and ultimately realising the United Nations' Sustainable Development Goals (SDGs) (e.g. Biewald et al., 2015; Donges et al., 2017a,b). A recent study identified a number of caveats, with potential implications for policy advice, if the dynamic feedbacks between changes in the biophysical Earth system and human system forcings are handled inconsistently or excluded altogether,

pointing to a "dark region of the uncertainty space" (Thornton et al., 2017). Similarly, using a model that explicitly includes feedbacks between human behaviour (not only their economic activities) and a changing environment, Beckage et al. (2018) found behavioural uncertainties to be as high as the uncertainties in modelling the physical climate system.

We consider the cultural component of the Earth system as important as biogeophysical, economic and technological components. Generally speaking, there is a need to account for the unique capability of humans (in contrast to other living

beings such as the aforementioned sheep) to act as a self-conscious force with foresight skills. Schellnhuber (1999) calls this collective cognitive capacity of humanity an immaterial, metaphysical "global subject"[1], which on the one hand happened to have "conquered our planet" (i.e. humans having appropriated Earth's resources), but on the other hand is now also on a quest for a more sustainable future as expressed in international agreements such as those mentioned above. Thus, we like to emphasize – as also pointed out by Schellnhuber & Kropp (1998) – that the "global subject" is a manifold cultural

phenomenon with distributed regional patterns as well as different aesthetical, cosmological and symbolic dimensions that coexist, evolve over time, and are driven rather by personal purposes or intentions than by functional or political purposes (Claussen, 2001; Lucht and Pachauri, 2004; Ehrlich and Kennedy, 2005; Gerten, 2009; Hulme, 2011; Grundmann, 2016; Haff 2017). This "pluralism, ambiguity and fluidity of most cultural systems" (Magistro and Roncoli, 2001) has also been emphasized by anthropologists calling for consideration of the multiplicity of worldviews in climate and Earth system

science (Bang et al., 2007; Reid, 2010). Thus, as noted above, the (possibly still dominant) "conquest" mindset coexists with an unidentified number of other mindsets, for example the striving for more humanistic and holistic visions, such as expressed in the SDGs and in the concept of a safe operating space encircled by environmental planetary boundaries and wrapped around a social foundation of justice and equity (Raworth, 2017). A more specific example is water management, which has undergone a number of paradigm shifts in the recent past, while a new paradigm towards more sustainable

practices is currently unfolding that may replace the former utilitarian/technocratic one (Gleick, 2000; Gerten, 2008). Accordingly, a number of future developments of the "global subject" (mindsets, values, executive organs) could be anticipated; yet, simulating such dynamics is complicated and still immature (e.g. Schellnhuber, 1999; Pahl-Wostl et al.,

---

[1] Other authors refer to similar concepts such as the "noosphere" (Samson and Pitt, 1999) or the "sociosphere"; different from the term "anthroposphere" that regards humans as subordinate to the biosphere (Mauelshagen, 2014).

2008; Sivapalan et al., 2012; Oldfield 2016). This is particularly true when it comes to the formal modelling of deeper humanitarian, ethical and religious motives central for understanding and governing Earth system processes in the Anthropocene – which we focus on here.

In the following sections, we (1) explain how religion(s) can serve as markers for modelling culture, and that their usage as such markers can enrich Earth system research with the said deeper human dimension; (2) propose environmental value sets to capture aspects of religion in modelling the 'whole' Earth system integrating the biophysical and the human mental realm; and (3) provide initial research primers for quantitative assessments in this direction. For the sake of forestalling misunderstandings, we like to stress that we are not making any statements about the metaphysical truth of any religion; neither do we wish to imply anything regarding the existential significance of faith for worshippers. We also refrain – and this needs to be underscored – from considering any one religion as superior to others. Instead, we focus on the possible environmental impacts of religions as collective societal phenomena and tangible cultural forces. We emphatically differentiate between the profound meaning of faith regardless of creed, which is outside the purview of our investigation, and the aggregate practices and policies that are correlated with the political representation of distinct religious communities in different cultural geographies.

## 2 Capturing the Human Component: Religions as Exemplary Markers for Modelling Culture

Considering the aforementioned "dark region of the uncertainty space", we take the argument to another level and ask whether and how culture can be represented in Earth system analysis and models (which is under-researched especially in quantitative terms), and which fundamental differences such enhancements may make compared to conventional approaches. The first step of our argument is to show that religion is more than a contingent and peripheral element of human culture, but instead one of its core features – even in societies that passed through the Enlightenment, whose institutions are defined by a division of church and state, and whose citizens and political representatives identify themselves as secular. This step hinges on two interrelated considerations: to understand religion in a wide sense, and to use this inclusive understanding for conceiving religion as being easily identifiable, strongly coherent, and culturally central. We basically follow Bloch's (2008) evolutionary argument that religion is a unique aspect of both human social organisation and a key (neurological) adaptation aspect of modern humans, characterised by the unique "capacity to imagine other worlds". Another general feature is that religion builds a strong link between the biophysical world and human imagination (Kong, 2010), providing coherence, a sense of connectedness and meaning (Tuan, 1976), thus potentially fostering pro-social activism and large-scale cooperation (Norenzayan and Shariff, 2008; Perry et al., 2015; Purzycki et al., 2016a). However, it has to be noted that religion also has the potential to lead to anti-social attitudes (e.g. Allport, 1954; Anderson, 2015), and that pro-social and pro-environmental concern and behaviour is a complex issue involving a number of personal and social factors besides religion (Gifford and Nilsson, 2014).

Our contention, that religion is central for predicting collective behaviour and eventual policy even in secular societies, conceives of religion as collectively shared and historically resilient belief in something that is held sacred. This construes religious phenomena rather broadly, and makes our underlying conception vulnerable to an objection discussed below. There are conflicting definitions of religion, and a persistent intractable issue is its proper scope. Some conceive of religion as (1) belief in God, others as (2) belief in gods or supernatural beings, and yet others as (3) belief in what is sacred (Allston 1967, Smith et al. 1995). Each of these successively broader conceptions has its merits, and none of them is without problems, which however do not spill over into the question of generic modelling that concerns us here. The more inclusive the concept of religion used in modelling is, the more of the uncertainty space such models will be able to elucidate.

A brief run-down of the pros and cons of the three conceptions of religion follows. The first and traditional conception of religion takes it synonymous with belief in God (1). This is the traditional Western notion, and it equates it with the three monotheisms of Judaism, Christianity, and Islam, while it excludes polytheistic and animistic belief systems, such as the creeds of Antiquity, the global belief systems of indigenous people, and Daoist religion in present-day China and Taiwan. Since it captures only a Western subset of belief systems centred on a supernatural and monotheistic deity, this conception is empirically too narrow. This suggests, (2), to identify religion with belief in gods or supernatural beings in general. This covers both monotheistic and polytheistic-animistic creeds, and is therefore more comprehensive, but it still fails to capture a religion such as Buddhism, the core doctrine of which centres on a holy person, the Buddha, who is neither god nor supernatural being. For the sake of including, say, Theravada Buddhism in Southeast Asia, Mahayana Buddhism in China and Tibet, and Zen Buddhism in Japan, we accordingly need (3) to identify religion with belief in the sacred.

This broad understanding of religion covers virtually all manifestations of worship in the world. While such an inclusive conception is practical for modelling purposes, it is not free of theoretical problems. The key objection is that it covers too much, since belief in whatever is held sacred is bound to include political and personal ideologies including those with adherence to ideas for which there is no empirical evidence or which are contradicted by historical and current data. There does not seem to be an easy way to exclude pseudo-religions, secular religions, and ideologies from such a definition. Pseudo-religious, totalitarian cults of personality are not merely artefacts of the past, such as in Fascism, Stalinism, and Maoism, but also political phenomena today, such as the state 'religion' of Kimilsungism in North Korean *Juche* ideology or the 'Dear Lady' cult in Burma of Aung San Suu Kyi, state counsellor of Myanmar (Martin, 2004; Heller et al., 2004; Houtman, 2005). Not even a purely secular ideology such as US-style neoliberalism can be excluded from this broad understanding of religion, since it contains quasi-religious traits. Examples are an unconditional faith in supply-side economics, a fundamental belief in the pursuit of self-interest, and an unconditional reverence of certain values, such as privatization and economic growth.

However, while this issue needs to be duly noted, it does not pose liabilities for modelling purposes. Some contemporary religious scholars, furthermore, have moved away from the traditionally narrow and 'essentialist' conceptions of religion (in that any and all religions share an essential belief in the supernatural), and favour instead a broader and heuristically more

useful "operational 'family-resemblance' approach," which seeks to illuminate religious experience and religion-resembling phenomena without drawing clear boundaries between what counts as religion and what does not, and which leads to "insights about the affective or spiritual dimensions of human perception and behaviour" (Saler 1993, Taylor 2016). For casting light on the aforementioned "dark space" of modelling human culture, what matters is to identify collectively shared, intensely held, and consistently implemented value sets, which are representative of certain cultures or hegemonic strata in societies, so as to predict large-scale behaviour of environmental relevance. Whether such value sets are those of traditional religions, such as Roman Catholicism; non-western creeds, such as Daoism and Buddhism; or even of highly problematic pseudo-religions, political cults, and secular economic ideologies, is immaterial.

In light of this broadly conceived understanding of 'religion,' we propose to employ religion as a marker for culture because of the following considerations:

(1) *Religions are central to cultures.* Without exception, religions constitute collective cultural identities, and they continue to exert hegemonic influence on what counts as the mainstream or the 'establishment' in many societies (e.g. for the US, *cf.* Harris, 1994; Layman, 1997; Baumgartner et al., 2008; Driskell et al., 2008). The consideration of the nexus of religion and establishment matters, because it is the latter that makes policies, in developed and developing countries alike. To be majority Sunni-Islam, for example, is a dominant trait of the culture of a developing country such as Pakistan; to be majority Protestant-Christian is a dominant trait of a culture of a developed country such as the US. So, religion is a central component to culture and influences societal behaviour accordingly.

(2) *Religions are strongly coherent.* Religious communities can be considered to be united by a shared set of outlooks, narratives, and values including environmental values (Hope et al., 2014). Sometimes these core doctrines remain invariant even if religions differentiate into denominations, as for instance environmentally relevant values in the Buddhist creeds across its variations Theravada, Mahayana, Chan, and Zen (all of which share environmental values through the so-called Four Noble Truths and problematize environmental impacts in their common earliest sources, the suttas of the Pali canon; *cf.* Daniels 2010). At other times, core doctrines fragment along the lines of such differentiation, as for instance in Christianity the opposition of Roman Catholicism and Evangelical Protestantism over the meaning of the natural creation in general and the import of anthropogenic climate change in particular (Bergmann & Gerten 2010). But in case of such fragmentation, the respective preponderance of core doctrines is also spatially separate, which poses little difficulty to modelling (e.g., Roman Catholicism represents cultural normalcy in Latin America and Quebec, while Evangelical Protestantism informs the establishment in the US). An illustration of normative coherence in spatial clusters is the relation of Christian communities in the Americas to climate change. In Latin American nations such as Mexico, in which 81% of the population describe themselves as Catholic and 9% as Protestant, an overwhelming majority of the population (82%) regards climate change as a "very serious" problem; in the US, with 20% Catholics but 40% Protestants, only a minority (40%) regards it as "very serious" (Evans & Zechmeister, 2018; Bell & Sahgal, 2014; Smith & Cooperman, 2015). Such normative coherence and spatial clustering makes modelling of human behaviour in discrete cultural domains feasible.

(3) Lastly, *religions are easily identifiable*. The spatial distribution of religious value sets and core doctrines is not elusive. Ample and detailed data on religious affiliation in nation-states and demographic groups are available, which are highly reliable, easily accessible, and continuously updated. This facilitates the use of religion as a cultural marker for modelling human behaviour.

## 3 Environmental Value Sets of Religions

There is overwhelming argumentative and observational evidence that religion plays a more or less explicit, if highly ambivalent, role in the perception of nature and in tackling environmental problems (White, 1967; Taylor, 2008; Bergmann and Gerten, 2010; Gardner, 2010; Jenkins and Chapple, 2011; Gerten and Bergmann, 2013; Veldman et al., 2014; Northcott and Scott, 2014; Brunn, 2015). For instance, religions offer moral arguments and potentially can mobilise civil society in support of a transformation towards socio-ecological sustainability (Dasgupta and Ramanathan, 2014). Taylor (2004) even noticed an ongoing trend towards "greening" religions as they are becoming more concerned with the environment in both theory and practice, although this development is very heterogeneous among different religions, and religious prescriptions certainly may not translate into actual behaviour (Kong, 2010). Also, recent quantitative findings based on a systematic exploration of historical, social and ecological datasets corroborate the theory that a shared belief in moralising high gods (supernatural beings imagined to have created or to govern all reality and to support human morality) and/or a shared reverence for what is deemed sacred can improve a group's ability to cope with environmental distress (Botero et al. 2014). A qualitative model developed by Stern (2000) assigns religion some causality for environmentally relevant behaviour, although the immediate contexts in the everyday world may be more crucial factors. This 'masked' role of religion may be supported by the fact – thus Stern (2000) – that countries such as India or China, where long-standing religious traditions featuring stark pro-environmental tenets prevail, do not show strong records of environmental protection. Indeed, some data-based analyses suggest that while religious belief increases personal concern about climate change, its explanatory power is often found to be statistically insignificant. One explanation is that belief systems tend to be rather persistent in time and space, being inert even in the face of dramatic environmental changes – at least in the short term, i.e. at time scales for which most empirical studies are performed (Stepp et al., 2003) which, however, may miss slow, incremental cultural dynamics. While the separate, quantitative effect of religion is difficult to isolate in empirical models (Tjernström and Tietenberg, 2008), values – i.e. deeply held convictions about what is right and wrong – have been successfully formalised (in so-called Values–Beliefs–Norms frameworks) as a foundational cultural influence on environmental decision-making and behaviour (Dietz, 2013; Caldas et al., 2015). Still, there is a need to develop quantitative techniques necessary to consider cultural factors in long-term and macro-scale Earth system analyses.

As has been pointed out by Gardner (2010), about 80% of the current world population identify themselves as being religious, and this bears the potential of religion becoming a major factor in developing new cultures of sustainability based on their inherent ancient wisdom of how to live a fuller life. Building on Veldman et al. (2013), Bergmann (2017) has

pointed out that religions have a number of specific functions in global environmental and climate change, in particular the following:

1.  Religion can either motivate or hinder environmental activism, thus its role is ambivalent, and also highly dynamic in time and space.

2.  Religious arguments and organisations in support of more environmentally sustainable ways of living, and concerns about the status of the environment, are clearly growing, as are responses to such declarations – though there are proportionally few voices from Islam and Hinduism.

3.  A great number of institutional and educational resources are owned by religious institutions (educational institutions, international networks, land ownerships, etc.).

4.  Religion fosters social connectivity, thus it bears potential to create empathy for people also in distant places and in future generations, and for other living beings.

5.  Finally, there is now a "spatial turn" at least in Christian theology (in contrast to the previously dominating concept of time), which recognises the complexity, diversity and global connectedness of creation on our home planet.

All of these features are of potential interest to better understand the broader role of humans in the Earth system, to develop visions of a more sustainable future and, thereby, to inform existing and newly developed concepts of Earth system analysis and modelling. Of particular significance for modelling Earth systemic influences of religion is its ambivalent, if paradoxical role (point 1 above). While one and the same religion can either motivate or hinder environmental activism in the society at large, such ambivalent roles are not consistently retained. In some cases, the nature of the function depends on the religion in question; in others, it depends on the internal institutional variants of the denomination. Lastly, in any (variant of) religion there certainly are both proponents with more conservative or more liberal attitudes (regarding environmental matters), who may have more in common with like-minded people from other religions than with the mainstream in their own religion. Similarly, there is a potential political relevance of people drawing individual conclusions from religious ideas and doctrines, possibly for their own benefit. Such caveats have to be carefully considered when assigning specific value sets to religions or religious communities, as they may modify (enhance, diminish, or possibly even neutralize) the eventual effect on the environment. Samples should thus be carefully selected and defined in order to account for such intra-religious divergences and ambivalences and to avoid oversimplified assumptions.

In world religions such as *Daoism and Buddhism*, the central role of the environment is unequivocal. Daoism and Buddhism (whose religious practices blend into one another in the often syncretic rituals of their religious communities in China and Taiwan) are generally 'green' and play an unambiguously positive cultural role in teaching environmental awareness to the greater public. This includes honing a collective awareness of climate change and motivating a societal push towards sustainability (Miller, 2017). The Chinese Taoist Association (CTA), the organisation of Daoism in the

People's Republic overseen by the State Administration of Religious Affairs, issued in 2006 the Qinling Declaration that identifies environmental harmony as the condition for sustainability and as the highest aim of Daoists (CTA, 2006). In 2007, perhaps even more remarkably, the CTA enshrined the sage Laozi as "Daoist God of Ecological Protection" (生态保护神; cf. He 2007). With the public embrace of sustainability as an explicitly religious goal, the CTA serves as a political platform for environmentalism in China.

Cases that do not consistently retain the ambivalent role of religion include Christian denominations. Here, positive or negative environmental functions depend, in part, on the variants of the Christian doctrine. While there is a wide range of denominations, as well as mixed cases that exhibit the ambivalent role described by Bergmann (2017) in their specific congregations, it is useful to consider poles of the Christian spectrum for the purpose of exemplarily modelling dynamics and effects of contrasting environmental value sets.

One such pole is the *Roman Catholic Church*. For motivating climate activism, Roman Catholicism has emerged as a worldwide cultural force under the leadership of Pope Francis and his unprecedented encyclical *Laudato Si'* (2015, esp. §§ 238–244). The papal call to the faithful to fight against climate change and solve other global environmental problems is hoped to make a political difference at least in nation states with Roman-Catholic majorities with representation in their governments, whether this be in Italy, Spain or Portugal in Europe, or in Costa Rica, Bolivia, and Argentina in Latin America. In North America, the pope's leadership has incurred criticisms by conservative clergy (e.g. the 2017 letter to Francis by US Bishop Weinandy; *cf.* also Landrum & Lull, 2017). However, while pushback by US conservatives against religiously motivated climate activism represents current government policy, US criticisms of the pope's leadership happen to be a minority position not only among US Catholics but also in the general population: Francis's 2014–17 approval rating in the US as a whole has been close to 70%; among US Catholics, his approval rating rose from 81% in 2015 to 87% in 2017 (Allen, 2017). In Latin America, which houses 40% of the world's Catholics, support for the papacy is stable and includes broad approval for leadership in climate activism and social justice (Evans & Zechmeister, 2018; Encarnacíon, 2014), as illustrated by the popularity of Francis's 2018 visits with indigenous tribes in the Peruvian Amazon and the Chilean Andes (McElwee, 2018a). A suggested explanation for this broad regional approval is that Francis' leadership mirrors already strongly held values – that "his relationship with the region runs both ways" and that "Latin America has increasingly influenced Francis' papacy" (Encarnacíon, 2014). Criticisms in Latin America, if any, concern the fear that the papal leadership is insufficient; that "not enough is being done" to stop the destruction (Chauvin, 2018). Actual protests, as in Chile, concern other matters, such as Francis's controversial appointments of bishops implicated in sexual abuse cover-up (McElwee, 2018b).

An opposite pole in Christianity is *Protestantism in the US*, whose ministers and congregations, especially in the Evangelical or Born-Again variant of Protestantism, are overwhelmingly sceptical of the credibility of climate science (83% of Americans identify as Christians, 53% as Protestants, and 37% as Evangelical or Born-Again; ABCNEWS/Beliefnet, 2017). Striking, in the exceptional role of the US in global climate politics, is not only the Protestant majority and Evangelical plurality of American Christendom, but also the political implementation of Evangelical views by the current

executive branch of the government. US President Donald J. Trump identifies as a follower of the Pentecostal Christian televangelist Paula White, who delivered the invocation at his inauguration on January 20, 2017. White is senior pastor of the Evangelical megachurch New Destiny Christian Center in Orlando, Florida. US Vice-President Mike Pence converted from Catholicism to Born-Again Protestantism and worships at the evangelical megachurch College Park Church, Indianapolis, Indiana (NYT, 2016). Evangelical megachurches teach creationism, climate denial, and the so-called prosperity gospel. The teaching of creationism cultivates a general scepticism about science among worshippers. Climate denial sharpens this scepticism into disdain for, and hostility of, climate science. The prosperity gospel, which teaches that wealth is a sign of God's love, while poverty signifies God's justice, desensitises the faithful against climate justice and undermines charity to environmental refugees. Both President Trump and Vice-President Pence emphatically deny the reality of climate change and strongly oppose climate action, which is reflected in the executive's political appointments, as that of the administrator of the US Environmental Protection Agency, Scott Pruitt, a climate denier and born-again Evangelical.

These three examples – Daoism in China and Taiwan; Catholicism in Europe or Latin America; and Evangelicalism in the US – suggest possible avenues for modelling Earth systemic functions of religion, not least since specific religious interpretations co-determine global (environmental) policies through statements and actions at highest political level. The environmental value sets of religions matter for motivating or hindering climate activism, for shaping collective environmental behaviour, and for informing government climate policy, if the following conditions are met. First, to have a broad impact, the religion must be a mainstream position, which can take the form of the major indigenous religion, as with Daoism in China, or that of a demographic majority, as with Roman Catholicism in various Latin American nations, or that of an economically and politically influential plurality, as with Evangelicalism in the US. Second, environmental value sets of religions matter for collective environmentally relevant (e.g. climate-forcing) behaviour if the associated communities of faith enjoy political representation, which takes different forms depending on the political system. In secular democracies with proportional voting systems, such as in the EU and Latin America, religious pluralities suffice to shape climate policies. In flawed democracies with winner-takes-all electoral systems and stakeholder-manipulated ("gerrymandered") voting districts, as in the US, religion shapes climate policy through the enacted beliefs of elected officials. In autocratic regimes, such as in China, religion shapes climate policy if it is condoned by the central government. If these conditions are not met, or if 'secular religions' such as communism or neoliberalism much more directly shape environmental policies by having politically more influential spokespeople as adepts, it will certainly be difficult to disentangle a particular effect of religion. To our knowledge, a systematic, international scrutiny of ideological influences on environmental policy decisions, which would be a prerequisite for their solid modelling at global scale, is still lacking. But formalizing contrasting religious viewpoints and associated value sets in a (stylised) model – which accounts for different religious viewpoints and their explanatory power relative to other factors, based on well-grounded theories and data – is still valuable for starting modelling exercises in this direction. Possible model approaches are discussed in the following section.

## 4 Possible Modelling Avenues

### 4.1 General conceptualisation

Assuming, thus, that religions or religious groups can be considered globally relevant 'agents', the question of interest here is how their Earth systemic relevance can be represented in respective models. In the following we suggest possible pathways to do so: namely a) more or less straightforward extensions of existing Earth system models by religious components and feedbacks; and b) design of new model types that specifically represent religion dynamics as a part or example of overall socio-cultural dynamics. Finally we propose to humanitarian questions from religious viewpoints that can inform and guide model formulation, model applications and scenario building.

A widely used, simple and robust approach for modelling an impact (I) on the Earth system is to consider it as the multiplicative response to anthropogenic driving forces. Formally, following the first 'IPAT' proposition by Ehrlich & Holdren (1972), this functional relationship can be described as: population size (P) times the affluence or economic activity per person (A) times the environmental impact of technologies used for producing goods and services (T). Importantly, the T term encompasses not only technologies (physical infrastructure) but is to be conceived also as an aggregate of "social organisation, institutions, culture and all other factors affecting human impact on the environment other than population and affluence" (Dietz and Rosa, 1997) – similar to Schellnhuber's (1999) generic notion of a "global subject" (see Sect. 1). Such a parsimonious formulation offers an entry point for conceptualising and quantifying functions of religion in the Earth system – analogous to the representation of biophysical impacts of P, A and parts of T via statistical and mechanistic relationships as typical for existing models.

However, the variety (and dynamics) of religions and their regional clustering obviously require representation in a geographically explicit manner, a prerequisite of which are appropriate databases. Thus, a mapping of diverse convictions and particularly the (more or less direct) influence of their bearers on real-world implementation, or hindrance, of environmental policies could help to understand the reasons and dynamics of (non)sustainable pathways of societies. While statistical databases with information about general religious features exist (e.g. the World Religion Database, http://www.worldreligiondatabase.org; data by the Pew Research Center, http://www.pewresearch.org/download-datasets; data collections from more specific projects such as Watts et al., 2015 and Purzycki et al., 2016b; or databases on wider cultural features of societies such as in Kirby et al., 2016) from which relevant information can be extracted (such as maps of the distribution and intensity of religious beliefs). However, "systematic mapping of the environmental function of religion" (Bergmann, 2017) remains a desiderate (also see Schimel et al., 2015). This is especially true regarding structured quantitative information on the influence of religion on environmental actions of societies/decision-makers, or more concretely, for instance, the interrelatedness of religious views, land-use, diet composition and climate change (see below).

## 4.2 Representing religion in existing biophysical Earth system models

Notwithstanding current data shortcomings, we can chart ideas for the above modelling variant a), i.e. inclusion of empirical relations between religious attitudes and practices and environmental processes into existing biophysical Earth system models (and also Integrated Assessment Models or the like). The analytical lens for the following examples is defined by climate change and planetary boundaries, i.e. focused on better representing interactions between human life and the fundamental physical limits of Earth to sustain modern human civilisation. Our premise is that humans mutually interact with their biophysical environment: they shape the environment, both deliberately and implicitly, and conversely their ways of life are being shaped by nature (in the short term by the provision of living requirements and by tangible adverse impacts such as floods; in the long term by setting the fundamental limits to human life). Religion can play a role in either of these directions.

Regarding the shaping of environments by humans, several pathways can be distinguished, one of which is direct intervention with landscapes, water and nutrient cycles through agriculture and forestry (besides processes such as urbanisation, river regulation or resource extraction for construction, an influence of religion on which is less amenable for analysis). As for agriculture, both cropping and livestock cultivation practices and demand for food (defined mainly by diet composition) are decisive for the human feedback to climate and the Earth system – in which religion and associated value sets may play a vital role. While in industrialised agriculture soils and ecosystems are primarily considered mere resources for production, traditional practices, especially if part of a religious/spiritual worldview that demands respect for planet Earth, tend to embrace a more holistic view. One of many examples is the Pachamama cult in South America, which regards Earth as a sanctity that merits adequate treatment and minimal disturbance (Sampietro Vattuone et al., 2009) – in contrast to mere techno-economic standpoints. Analogously, it can be asked inasmuch the high deforestation rates in past decades have a root in the modern treatment of forests as a mere object, in contrast to ancient views of trees as deities or subjects with dignity, which still exist in traditional societies and e.g. in Japanese Shintoism (White, 1967; Northcott, 2010). More direct influences of religion are obvious for diet composition: the still substantial share of vegetarian diets in India (in stark contrast to e.g. the US), as well as the renouncing of pork meat by Muslim and Jewish religion, are large-scale examples of how religious convictions may codetermine global food demand and thus the biophysical impact of agriculture. To our knowledge, a systematic, quantitative and intercomparative study of whether societies with a dominance of specific religions have an overall lower environmental footprint is still lacking. If established, findings from such studies could provide the basis for enhancing Earth system models by representing different ways of agriculture or forestry and analysing their subsequent impact on the Earth system as influenced by distinct religious viewpoints. Model enhancements of this type can learn from work on inclusion of altruism and other 'soft' cultural dimensions in economic models, which eventually better reproduce real-world social dynamics than models driven by purely economic incentives (Henrich et al., 2001; Fehr and Gaechter, 2002).

Regarding the shaping of human life by environmental conditions, climate change impacts (particularly extreme events

such as droughts, heatwaves, floods) and adaptation will play an increasing role in the near future. Religion may play a central role in how such events are perceived and how societies cope with them (Palmer and Smith, 2014). On the one hand, people may interpret them as a punishment of God for sinful living – a fatalist attitude which may eventually make them restrain from adequate mitigation and adaptation actions (see e.g. Gerten, 2010). On the other hand, catastrophic events may in some instances trigger increased social coherence when it comes to coping with impacts – though there are counter-examples such as the (unsuccessful) attempts of US Evangelicals to blame natural hazards such as hurricane Harvey on homosexuality (e.g. Independent, 2017). Such differences in human attitudes and potentials, influenced by religious convictions and rituals, may provide avenues for improved representations of societal impacts and adaptation in local- to global-scale models and scenarios.

Finally, human impacts on the environment certainly depend on population size (see the IPAT model), rendering it important to quantify the potential influence of religion on population growth. Besides the traditional stance against contraception in Catholic teaching, Iran can serve as a marked example, where population was growing by ~3% yr$^{-1}$ after the Islamic Revolution in 1979. After a decade of politically fostered growth, politicoreligious leaders reverted their views and promoted family planning, eventually achieving much lower growth rates of ~1% (Aghajanian, 1995; Hoodfar and Assadpour, 2000). Though the religious framing of both growth and stallment policy is by far not the only influence, it nevertheless demonstrates a prominent role of religion in this context since political and religious leaders are strongly intertwined in Iran.

A critical question to be addressed by respective data analyses and model applications (see also Sect. 4.4) is whether an influence of religion on past (macro-historic) patterns of population, dietary choice, land use, ecological footprints etc. can be solidly quantified, what dynamic changes in such relationships have occurred (in the recent past), and whether religion may still be a discernible factor in future developments: For example, the influence of religious traditions on collective choices on reproduction and practices to meet the demand for food, freshwater and living space may (have) vanish(ed), as government-funded education or support for fertility control as well as individual affluence may increase in importance. It has to be noted, though, that any government is not *per se* devoid of religious conviction, or even an antagonist to religion – counter-examples are Turkey or Iran, where Islam has a large role in the shaping or justification of policies.

## 4.3 Developing specific socio-cultural models for aspects of religion

Under modelling variant b), a number of different model types can be subsumed: namely any approach designed to explicitly study social dynamics influenced by religious agency and relevant at macro-scale. A starting point can be agent-based models that take into account human adaptation and learning, to better represent the (networked) dynamics of different social (here, religious) groups and actors (Farmer & Foley 2009, Donges et al. 2018; and see Sect. 1). As elaborated in Sect. 3, a basis for such modelling is to assign distinct lifestyles and environmental footprints – i.e. different socio-metabolic classes

(Fischer-Kowalski 2011) – to specific religions or religious groups (which either concentrate and operate in specific regions and/or affect outcomes at supraregional up to global scale), who are ascribed some competence for sustainability transition, and whose dynamics and Earth systemic imprints can be modelled.

An example from the US may serve as an illustration, namely two Christian groups who have contrasting views on how to deal with natural resources: While the Amish people in Pennsylvania, Ohio and Indiana deliberately choose to follow a low-energy-intensive lifestyle conforming with their ethical convictions, several prominent Evangelical preachers and politicians fiercely vow for full exploitation of natural (fossil) resources as a gift of God, usually in conjunction with climate change denial (e.g. Kearns 2013; see Sect. 3). Especially note the recent withdrawal from the Paris Agreement and the reappraisal of the fossil fuel industry by the Republican Party under Trump, which echoes the latter ideas. While a global biophysical effect of this position change by the current US administration can be calculated (i.e. the additional global warming resulting from the lowered emissions reduction pledges, see climateactiontracker.org), it is ultimately worth to analyse the deeper socio-cultural motivations of such retrograde, disruptive environmental politics, and the role of religious stances therein. Models able to formalise such knowledge could then be used to explore Earth systemic consequences of a (stylised) spectrum of such decisions, ideally through data exchange with, or direct coupling to, conventional Earth system models (Donges et al., 2018). As a counter-example, the *Laudato Sí*, published shortly before the COP21, is expected to provide a certain stimulus for a democratic cultural turn towards more sustainable and sufficient ways of living (Brulle and Antonio, 2015): This points to the possibility that declarations by religious leaders, or religious groups as civil society actors, may influence the formation and dynamics of new pro-environmental opinions and coalitions across countries and cultures (Bedford-Strohm, 2008). Modelling such dynamics would be also important to enrich or complement research on 'social tipping points' – bifurcative transitions in societies (Bentley et al., 2014) – which are hardly predictable and currently ill-represented in Earth system models despite being likely to produce crucial nonlinear imprints onto the Earth system. However, religious declarations or movement texts are not to be overrated as an authoritative statement that will produce widespread changes in mindsets or immediate real-world action; their impact may be far weaker than expected, or they can even be strongly resisted (Landrum et al., 2017). Actually, Landrum and Lull (2017) observed that key messages of the *Laudato Si'* did not resonate well with the morals and convictions of conservatives in the US, thus having little influence on the decision to pull out of the Paris Agreement (which has been strongly criticised not only by the Pope and Catholic leaders but also by interfaith movements, e.g. parliamentofreligions.org/parliament/interfaith-climate-action-new/response-trump-withdrawal-paris-agreements).

Ideally, modelling such dynamics captures the coevolution of religious views/practices and (actual or anticipated) environmental changes, that is, how the physical–mental–physical feedback dynamically evolves over time; such approaches have been successfully tested, for example, in socio-hydrological and integrated assessment modelling case studies (Sivapalan et al., 2012; Kelly et al., 2013; Blair and Buytaert, 2016). Recently developed model approaches, which are able to quantitatively describe dynamics of macro-scale social relationships and transmission of cultural changes as still

influenced by ancient (linguistic and also religious) ties, appear to be promising as well (Matthews et al., 2016). Existing approaches range from simple generic models to context- and data-driven models, and they are generally confronted with the need to balance sufficient process representation and parsimony (see Garcia et al., 2016). Another example model type is the inclusion of shades of religious ascriptions into game-theoretical considerations of e.g. the climate conferences, to better

predict their outcome or to elucidate how defection can be avoided (Sprinz et al., 2016). Furthermore, response options to an imminent, local crisis of food security could be constructed or enhanced by integrating prevalent religious convictions into intervention simulations to avoid further social unrest (Hendriks, 2015).

The type, complexity and parameterisation of any such model depends on the research question to be studied (see following section for ideas). One of many possible examples for a model approach to explore an interaction between religion on the

natural Earth system – here, following Norgaard's (1984) definition of a quasi-coevolutionary relationship whereby human and environmental systems mutually interact in such a manner that they impact one another's developmental trajectory – could consist of the following elements: 1) a representation of a religious community (or rather the collective proponents of a religion if a global effect is to be analysed) as a dynamic agent who pursues actions related to a specified environmental value set such as the aim to stay within planetary environmental guardrails; 2) a dynamic and ideally spatially distributed

representation of the biophysical or biogeochemical processes under investigation, such as one or more of the planetary boundaries (which can be represented in a simple way like in Heck et al., 2016); and 3) a coupling mechanism that enables data exchange between these modules at specific simulation time steps in order to represent feedbacks between the two. Broadly speaking, one could initially force this model with the observation that several planetary boundaries are currently transgressed or near transgression. This could translate into the assumption that a religious community or organisation under

consideration – collectively or partly – fosters and/or pursues activities to use Earth's resources more sustainably (possibly aided by communication channels such as social media with networked effects across the globe). The cumulative effect of such activities could then be interpreted by the biophysical module in terms of the extent to which a relaxation of the pressure on the planetary boundaries occurs. Such a model could also be used to contrast the environmental outcome if another religion or worldview with rather unsustainable attitudes (e.g. retrograde environmental policies favoured by US

Evangelicals) dominate. Eventually, the joint dynamics of diverging mentalities/policies can be combined in a more complex approach. Note our comments in previous sections that a heuristic model in which specific value sets are assigned to specific religions is a simplistic, stylised way to learn about possible (nonlinear) system dynamics (also see Heitzig et al., 2018). In an ideal case a multifactorial model is developed that accounts for intrareligious differences in attitudes and activities (based on empirical observations), and that also represents how institutions and norms affect collective behaviour at all, with

religious value sets implemented via the medium of governments.

## 4.4 Guiding research questions

In this section we propose some research questions from specifically religious viewpoints that might be addressed by Earth system models and that otherwise may be overlooked despite being highly relevant in terms of which futures are perceived in the modelling process. In other words, religious views on the very role of humans on this planet and their future visions may differ from, if not oppose (but ideally complement), economic and/or technocratic optimisation paradigms that guide many current model structures and applications (see Sect. 1). Indeed, religious – and ethical and moral – questions belong to Earth system analysis *sensu latu*. Earth system models could be used to imagine, and represent, alternative worlds beyond technocratic-economic pathways, i.e. "fundamentally different visions of the good life" using a "compelling narrative of transformative social change" (Brulle and Antonio, 2015) while highlighting that "mankind is not fatally trapped in an inescapable tragedy of the global commons" (Edenhofer et al., 2015). This gives room for a new sort of questions that could be addressed quantitatively by enhanced or newly developed Earth system models, also contributing to the development of narratives about possible (mental, cultural) futures in the IPCC's Shared Socio-Economic Pathways (SSPs; O'Neill et al., 2017) – like, for example:

- Do religious practices and theories have explanatory power for major Earth system dynamics and trends, such as anthropogenic climate change or meat consumption?

- Can religions make a noticeable contribution to "reconnecting [humans] to the biosphere" (Folke et al., 2011)? Which visions of more holistic lifestyles do they offer that in the long run are more sustainable than economic-technologic paradigms?

- What are differential impacts of utilitarian *vs.* theistic worldviews on the Earth system? Analogously: what, if any, are the differential impacts of monotheistic *vs.* animistic-polytheistic religions on the Earth system, and what, if any, are the differential impacts within monotheism between more humanistic and less humanistic communities of faith on the same?

- Do differences in "cultural logic" – influenced by religious convictions – actually result in different environmental policy changes, or do they converge even though underlying motivations are different (see Zhu and Jesiek, 2014)?

- If key recommendations in the *Laudato Si'* or, in a broader sense, in Prince Charles' (2010) approach, i.e. transitions toward more sufficient lifestyles, were taken serious across cultures, would it then be easier to achieve the SDGs while staying within planetary boundaries?

- Do utopian narratives of more sustainable futures produce mental feedbacks with current societies, triggering social tipping points that help direct societies onto pathways that lead to this very future?

- If mitigation and (technological) adaptation with respect to global climate change and other Earth system changes fail, how could religious beliefs, or (inter)religious aid coalitions, help to cope with resulting suffering and death?

- Is the formation of such coalitions facilitated by (dystopian or utopian) projections of climate and Earth system change, or only in the aftermath of actual catastrophic events? What exactly triggers (and has triggered in the past) the formation of such movements – is it the moral responsibility? More generally, would religious value sets help or hinder timely recognition of the fact that we are currently on an unsustainable path (*cf.* Tainter, 2006)?

- How much suffering is acceptable at all – in other words, where to position (deduced from religious points of view) 'moral' tipping points of Earth system change? Are they already reached due to the impacts of a mean global warming by 1.5 or 2 degrees (perhaps because poor people are already strongly affected then)? And are the collective impacts of other environmental changes, such as the widespread water scarcity and biodiversity loss, already way beyond a morally acceptable level?

- What (biophysical, political, social) effect would it have if religious communities around the world declared that all (land, water) resources on the territories they own were to be protected from human appropriation? What if they invested large amounts of money in environmentally sustainable practices? What politicoreligious and social dynamics will unfold when more and more sacred places will be threatened e.g. through sea level rise?

- How would answers to these questions look like from the perspective of different religions?

Ultimately, how can interdisciplinary research between the humanities and the natural sciences, the dialogue between science and religion, and also interfaith dialogue, be put forward so as to enrich Earth system analysis as a means to foresee more sustainable futures of the Anthropocene – without attempting to enforce a quantification of essentially qualitative (philosophical, moral, etc.) aspects?

**5 Conclusions**

This article enters fairly new and probably unconventional grounds for most modellers. Yet, in our view, it addresses issues of high relevance when aiming for an improved understanding of the increasingly pressurised Earth system and its possible trajectories into the future: how to enrich, newly build, or inform, Earth system models by integrating cultural components – and specifically functions of religion as they represent core human values, morals, empathic capabilities and future visions? Due to the dearth of existing approaches, we could only sketch how a respective modelling landscape might look like,

cornerstones of which can be summarised in the following recommendations for next steps of research in this direction:

(1) Compile an empirical cross-cultural database (with qualitative meta-information and as part of the larger quest for socioeconomic data in modelling), targeted to quantitatively represent religious factors of environmental relevance and serving as a fundament for – still lacking – systematic scrutiny of religious functions in the Earth system. (2) Identify key feedbacks between activities of religious groups/actors and global environmental changes, and represent them in existing

complex Earth system models to distil, and possibly project into the future, their particular effect. (3) Construct new parsimonious models – possibly linked to more complex Earth system and Integrated Assessment Models – specially designed to analyse how religious theory and practice coevolve over time with the global environment. (4) Identify research

question of macroscale humanitarian relevance that can be addressed by such enhanced models, and provide respective insights in pilot studies – including further conceptual development of here provided initial ideas.

On a final note, we stress that it remains to be studied comprehensively whether any particular religion might be superior regarding its potentials to support a transition toward a more sustainable world, as has been occasionally suggested (Schönfeld 2010; Minton et al., 2015). Any scientific inquiry of religious attitudes and their environmental relevance should be neutral and free of subjective statements. This is especially true since the role of religious thought and practice in global environmental change is highly ambivalent and complex, with progressive and regressive tendencies at work, and with ample evidence for morals and practices that rather undermine than support sustainability efforts or, for example, result in fatalism or dogmatism. This ambivalence itself merits neutral, systematic and intercomparative analysis in an Earth system context, and it only reaffirms the research gap asserted here: to take at face value the 'Anthropocene' notion and more fundamentally understand, and possibly (though not necessarily) quantify, the immaterial dimension of humanity's imprint on our planet – thereby breathing life into otherwise 'cold' quantitative analyses and narratives.

**Acknowledgments:** We thank the Editor, John Finnigan, Anne Marie Grisogono, and an anonymous reviewer for their constructive comments; and Jonathan Donges for discussions on the topic of this paper.

**Competing interests:** The authors declare that they have no conflict of interest.

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
