# Peer review of "On deeper human dimensions in Earth system analysis and modelling"

_Earth System Dynamics, 2017_

## Referee Comment (RC1) · Anonymous Referee #1 · 16 Jan 2018

This is an interesting and important area that needs to be included in Earth System modelling. The authors have given a good overview of religions and how the belief systems can influence the way humans are using and impacting the earth system, and the biosphere in particular. I agree with the three approaches suggested to further this area, but I think the last one – c) distillation of deeper humanitarian questions from religious viewpoints that can inform and guide model applications and scenario building — needs to be done at the start or in conjunction with the others. This relates to the main area that needs expansion: The authors refer several times to how religions are interpreted, and the extent to which this makes the inclusion of religion in modelling worthwhile. They mention it again in the last paragraph. The use of religion to achieve personal or group aims is widely acknowledged. It's not only USA evangelicalism. All

religions interpret the divine messages to suit their purposes. Does this enhance or modify the effect the group is having on the environment? Or does it just make (that) religion irrelevant? The paper needs expansion of this area, with some indication of how it should be approached from a research perspective.

I note a number of typos, but have not recorded them. The paper needs editing.
* * *

---

## Referee Comment (RC2) · A. M. Grisogono (Referee) · 8 Feb 2018

The paper identifies an important lacuna in the literature on earth system modelling and analysis, and proposes a brave approach for addressing it objectively, building on existing knowledge bases and techniques.

It is necessarily a risky enterprise given the potential for strongly emotional and subjective reactions from fervent adherents of particular beliefs, whether through challenging the global impact of their values, or fuelling a 'blame game' that may further sharpen the divisive polarisation we already see in national and international politics.

Nevertheless, the authors are correct in pointing out the risks of overlooking the impacts of the geographical distributions and strengths of different religious value sets on

how human societies interact with environmental and climate change.

Of course, religion is not the only cultural determinant that influences the climate change relevant behaviours of social groups, and it would be good to see how to set the proposed research program into a wider socio-cultural context, which might ultimately provide a more comprehensive account of the complex trajectories we observe and wish to influence. My main concern is how these 'sacred cows' can be treated in a balanced and rational way with scientific integrity, without provoking a destructive backlash.

The text would benefit from some minor editing, as noted by the other referee.

---

## Editor Comment (EC1) · J. Finnigan (Editor) · 22 Mar 2018

This paper argues persuasively for the inclusion of important socio-cultural factors in earth system models, particularly in the IAMs used to design and test human policy responses to climate change mitigation and adaptation. It focusses on religion(s) as the primary shapers of human decision making at population/voting bloc level and teases apart the attitudes of different religions and sub sects to environmental protection and the use of earth's resources. The aims of the paper are threefold:

1. To explain how religion(s) can serve as markers for modern culture so that their inclusion can add a deeper human dimension to earth system modelling.

2. To propose environmental 'value sets' to capture aspects of religion in modelling the

whole earth system (understood here as the intersection of the social and biophysical worlds)

3. To provide some initial research primers for quantifying the required steps.

I think the paper succeeds quite well in approaching all three aims but leaves open a serious gap between the influence of religion on human attitudes and the way these attitudes come to be expressed in decision and policy making. I'll expand on this view below.

1. religion(s) as markers for modern culture

It makes sense to adopt the 3rd widest definition of religion, that is, as a belief in the sacred but I wonder if the definition of 'sacred' also needs to be wider? Communism was effectively a set of beliefs about the optimal way of organising social interaction and 'the means of production', which were received and largely unquestioned by generations in the eastern block post 1917. Capitalism and the primacy of the individual has held the same position in the USA through most of the 20th century and today. Economic and cultural nationalism (autarky) had a powerful and ultimately disastrous airing between the 2 world wars and seems to have reappeared on the world stage today.

More recently, a sub sect of capitalism, neo-liberal economics, has held sway in most Anglophone polities for the last 45 years. A feature of both neo-liberalism and communism has been the unquestioned adherence of its proponents to ideas for which there is no empirical evidence or which are contradicted by historical and current data (eg. trickle down economics). More important for the arguments in this paper is that these 'secular religions' directly shape policy whereas more traditional religious beliefs have to be filtered through a further step of electing representatives who profess those beliefs, at least in democracies.

Communism, at least in its soviet interpretation of Engel's dialectic materialism, had

disastrous environmental impacts. The untrammelled application of neo liberal economics today seems to be producing the same results but based on a totally different philosophy

**2. environmental value sets**

this is a useful survey of the attitudes of different major sacred (as opposed to secular) religions to the environment and to humanity's use of natural resources. However, I wonder if the authors' assessment of the role of religion is not more positive than is justified. For example, the papal encyclical 'Laudato Si' certainly said all the right things but it came from a Pope whose liberal views are strongly opposed by many powerful conservatives in the church hierarchy. The liberal-conservative battle in the catholic church is playing out on several fronts with cover ups of child sexual abuse and the role of women being major battle grounds but conservation and climate change is also far from accepted by conservative Catholics. The 'number two' in the Vatican hierarchy, Cardinal George Pell is a prominent climate denier and many other examples of powerful conservative figures who oppose Laudato Si can be found.

Hence, I do not wholly agree that the catholic church forms one pole of the Christian spectrum with regard to environmental and climate attitudes, evangelical Protestantism being the other. Rather it seems to me that conservatives in both catholic and protestant churches might have more in common with each other than with liberals in their own churches. Similarly, conservative Islam has many attitudes in common with catholic and protestant conservatives in areas like birth control and creationism so it is questionable quite how well the major religious divisions used in the paper map onto value sets.

There are a few other statement in this section of the ms that could be challenged. For example, that religion fosters social connectivity and so has the potential to create empathy for people in distant places and future generations. Religions are also powerful ways to define and exclude 'the other' so that adherents don't have to feel empathy.

There are of course ample examples from history but the demonization of the poor by US evangelical Prosperity Theology is a good current example of how a religion can be used to avoid empathy with the disadvantaged.

In summary, I found this section a good first attempt at identifying religions and value sets but the objections I raise above suggest that a subtler multi-factorial classification might be needed to operationalise this concept in any model.

3. Modelling avenues

Three possible pathways are considered: a) extensions of existing models; b) design of new model types; c) Distillation of deeper humanitarian questions from religious viewpoints that can inform and guide modelling. The authors suggest that religion can be considered a social technology in an IPAT formulation of environmental impact of society. This is unarguable but simply displaces the question of how to operationalise this in models.

a. extensions of existing models

The suggestion here is to adjust human impact on the environment using empirical data on the correlation between the religious value set and observed impact on the environment and also environmental feedback on human behaviour. Examples posited of the forward impact were the religious shaping of agro-forestry, pastoralism and land management practices; dietary preferences and population growth. The first four of these have certainly been important in the past but it might be argued that, in the future, satisfying the demand for food and living space will swamp religious traditions. Collective choices on reproduction, for example through cultural expectations for large families, clearly affect affect the population and thence impact through IPAT but government funded education or support for fertility control and simple per capita wealth probably has more ultimate influence on population.

Also, I found the idea that climate catastrophes would be blamed on supernatural displeasure and would also promote solidarity to be in the first instance unlikely and in the second optimistic. US evangelists tried to blame Hurricane Katrina on homosexuality without notably convincing many and at the same time Katrina apparently led to societal breakdown rather than solidarity. The main problem with this approach is that for the largest factors shaping human impact on the earth, religious values are mediated through political and economic processes.

Unfortunately, when we come to pathway b, there are no concrete examples of the kind of model the authors have in mind, other than a suggestion that attributes of agents in agent based models should include religious value sets that affect land use. As a result, it is difficult to see precisely what is envisaged here.

Pathway c is illustrated rather by sets of research questions that bring religion and ethics into questions of what future scenarios we might as a species or as separate populations find acceptable. This approach might fit quiet well with the development of the SSP framework for IPCC IAMs.

Summary

This paper casts a refreshingly unconventional eye on modelling earth system dynamics by including socio-cultural factors as embodied in religious beliefs and practices. It advances a series of assertions about the influence of religious value sets on human impact on the planet, many of which can be challenged, or at least to which significant exceptions can be found. I believe its main weaknesses are threefold. First, the religious value sets are not as clear cut as the authors suggest and a more multi factorial approach may be needed.

Second, when it comes to model pathway b, the construction of new models, the paper needs to suggest some concrete examples of the way the authors think this would work. There is a disconnect between some high level statements about how land use is land use shaped by religious practice, which may have been relevant once but are probably increasingly irrelevant as we head towards feeding and housing 10-11Bn

humans. More specificity here would allow readers to grapple better with the paper.

Third, one can make a strong argument that the principal way that religious value sets are implemented is via the medium of governments as they respond to their polities. The paper makes this point well early on but when it comes to describing new modelling paths, this seems to be lost. In other words, a more general formulation is required of how institutions and norms affect collective behaviour. These institutions and norms include religious (sacred and secular) value sets and thereby govern impact on the biosphere and collective reaction to biospheric changes. Trying to shortcut this level of understanding as suggested in modelling path a seems to invite false correlations and parameterisations.

Finally, I wonder if the authors want to comment on how religious value sets affect the way society reacts to other factors that operate outside its conscious understanding. I'm thinking of factors such as those described by Tainter in his many treatises on the collapse of Complex Societies (eg. Tainter, 2006 and refs therein) once the returns to complexity start to diminish. Past societies found various explanations for what must have seemed like inexorable negative forces out of their control. Would current religious value sets help or hinder our early appreciation of the fact we are on an unsustainable path?

Tainter, J. A. (2006) Social Complexity and Sustainability. Ecological Complexity. 3. 91-103

---

## Author Comment (AC1) · 22 Mar 2018

We are grateful for the referee's overall positive assessment of our paper and the constructive comments, to which we intend to respond as follows.

Thank you for pointing to the paper-structural issue that our point c needs to be done at the start or in conjunction with the others. We agree that point c) is not at the same level than the two other points, as it does not provide a model avenue as such but rather suggestions for research questions that can be addressed by models portrayed under points a) and b). We will therefore consider your suggestion to present the proposed research avenues prior to the model approaches, which appears to be a better structure for these sections of the paper (research questions first, methods thereafter).

[Figure]

You are right that some elaboration of the aspect you mention – whether it may be important that people draw their individual conclusions from religious ideas and doctrines (possibly for their own benefit) – is helpful. We will thus include a paragraph on this strong ambivalence, ideally in the section where we discuss that religious theory and practice may diverge in terms of their ultimate environmental impact.

Also, for the sake of forestalling misunderstandings, we intend to add a differentiation to the beginning of the paper regarding the aspects of religion that we wish to investigate: We are not making any statements about the metaphysical truth of any religion; and neither do we wish to imply anything regarding the existential significance of faith for worshippers. Instead, we focus on the possible environmental impacts of religions as collective societal phenomena and tangible cultural forces. We emphatically differentiate between the profound meaning of faith regardless of creed, which is outside the purview of our investigation, and the aggregate practices and policies that are correlated with the political representation of distinct religious communities in different cultural geographies.

Finally, we will carefully check the revised version for any typos and other editorial and stylistic improvements.

---

## Author Comment (AC2) · 22 Mar 2018

Thank you for your overall positive assessment of our paper and the helpful critical comments.

Indeed, as you note, we tried to address the possible and diverse roles of religion for global sustainability (and models of the earth system) as objective and careful as possible, pointing to both the positive and the negative sides – see our remarks in the final paragraph. We will consider pointing this out also in the Introduction in the revised version. Regarding your comment that such a paper and follow-up research "is necessarily a risky enterprise given the potential for strongly emotional and subjective reactions from fervent adherents of particular beliefs". For the sake of forestalling

misunderstandings, we intend to add a differentiation to the beginning of the paper regarding the aspects of religion that we wish to investigate: it is not about statements on the metaphysical 'truth' of any religion; we rather focus on the possible environmental impacts of religions as collective societal phenomena and tangible cultural forces. Also see our intended response to the other Referee's comments.

It is a valuable suggestion to improve the framing of our paper and the suggested research field (religion in earth system analysis) regarding existing literature on wider socio-cultural research. For the revision we aim to briefly review existing key literature – while focussing on the other aspect you mention, i.e. that scientific inquiry of religious attitudes and their environmental relevance should be neutral and free of subjective statements (also see the above comment). In this context we will consider including a statement on the difficulty of achieving this very balance, especially in practice. Also see our intended response to the Editor's remarks, for which we will point more explicitly to the ambivalence of religion.

Finally, we will carefully edit the revised version.

---

## Author Comment (AC3) · 28 Mar 2018

This paper argues persuasively for the inclusion of important socio-cultural factors in earth system models, particularly in the IAMs used to design and test human policy responses to climate change mitigation and adaptation. It focusses on religion(s) as the primary shapers of human decision making at population/voting bloc level and teases apart the attitudes of different religions and sub sects to environmental protection and the use of earth's resources. The aims of the paper are threefold: 1. To explain how religion(s) can serve as markers for modern culture so that their inclusion can add a deeper human dimension to earth system modelling. 2. To propose environmental 'value sets' to capture aspects of religion in modelling the whole earth system (understood here as the intersection of the social and biophysical worlds). 3. To provide

some initial research primers for quantifying the required steps. I think the paper succeeds quite well in approaching all three aims but leaves open a serious gap between the influence of religion on human attitudes and the way these attitudes come to be expressed in decision and policy making. I'll expand on this view below.

Thank you for your elaborate and thoughtful comments, to which we intend to respond as written below.

1. religion(s) as markers for modern culture It makes sense to adopt the 3rd widest definition of religion, that is, as a belief in the sacred but I wonder if the definition of 'sacred' also needs to be wider? Communism was effectively a set of beliefs about the optimal way of organising social interaction and 'the means of production', which were received and largely unquestioned by generations in the eastern bloc post 1917. Capitalism and the primacy of the individual has held the same position in the USA through most of the 20th century and today. Economic and cultural nationalism (autarky) had a powerful and ultimately disastrous airing between the 2 world wars and seems to have reappeared on the world stage today. More recently, a sub sect of capitalism, neoliberal economics, has held sway in most Anglophone polities for the last 45 years. A feature of both neo-liberalism and communism has been the unquestioned adherence of its proponents to ideas for which there is no empirical evidence or which are contradicted by historical and current data (eg. trickle down economics). More important for the arguments in this paper is that these 'secular religions' directly shape policy whereas more traditional religious beliefs have to be filtered through a further step of electing representatives who profess those beliefs, at least in democracies. Communism, at least in its soviet interpretation of Engel's dialectic materialism, had disastrous environmental impacts. The untrammelled application of neo liberal economics today seems to be producing the same results but based on a totally different philosophy.

Thanks for acknowledging our wider definition of 'religion' and for pointing to the fact that e.g. communist and neoliberalist positions may similarly be grounded in convictions/philosophies that do not have empirical evidence (and probably have even more

directly an effect on the environment than religious convictions). In the revision, when defining the 'sacred', we will consider this aspect and extend our statements on totalitarian cults, pointing out that such (neoliberal) unreason may indeed be akin to religious dogmatism.

2. environmental value sets this is a useful survey of the attitudes of different major sacred (as opposed to secular) religions to the environment and to humanity's use of natural resources. However, I wonder if the authors' assessment of the role of religion is not more positive than is justified. For example, the papal encyclical 'Laudato Si' certainly said all the right things but it came from a Pope whose liberal views are strongly opposed by many powerful conservatives in the church hierarchy. The liberal-conservative battle in the catholic church is playing out on several fronts with cover ups of child sexual abuse and the role of women being major battle grounds but conservation and climate change is also far from accepted by conservative Catholics. The 'number two' in the Vatican hierarchy, Cardinal George Pell is a prominent climate denier and many other examples of powerful conservative figures who oppose Laudato Si can be found. Hence, I do not wholly agree that the catholic church forms one pole of the Christian spectrum with regard to environmental and climate attitudes, evangelical Protestantism being the other. Rather it seems to me that conservatives in both catholic and protestant churches might have more in common with each other than with liberals in their own churches. Similarly, conservative Islam has many attitudes in common with catholic and protestant conservatives in areas like birth control and creationism so it is questionable quite how well the major religious divisions used in the paper map onto value sets.

Thanks for pointing so clearly to inner-religious differences, which certainly cannot be neglected as demonstrated by the examples you mention. The case of Laudato Si' (its criticism from Catholics) became more pronounced and public only recently, and we will try to find new references to support any statements that can be made about this criticism. In general, in the revision we will elaborate more on the many ambivalences

when it comes to religious influences on environmental issues, and we will point to the fact that political leaders (even in strongly Catholic counties, for example) may not pick up any religious views in their decisions – indicating that comprehensive and neutral research on such topics is duly necessary.

There are a few other statement in this section of the ms that could be challenged. For example, that religion fosters social connectivity and so has the potential to create empathy for people in distant places and future generations. Religions are also powerful ways to define and exclude 'the other' so that adherents don't have to feel empathy. There are of course ample examples from history but the demonization of the poor by US evangelical Prosperity Theology is a good current example of how a religion can be used to avoid empathy with the disadvantaged. In summary, I found this section a good first attempt at identifying religions and value sets but the objections I raise above suggest that a subtler multi-factorial classification might be needed to operationalise this concept in any model.

See previous comment: in the discussion paper we sometimes pointed to the different functions and ambivalences of religion(s), but we will now better work out the complexity of the issue (as regards environmental and earth system science).

3. Modelling avenues Three possible pathways are considered: a) extensions of existing models; b) design of new model types; c) Distillation of deeper humanitarian questions from religious viewpoints that can inform and guide modelling. The authors suggest that religion can be considered a social technology in an IPAT formulation of environmental impact of society. This is unarguable but simply displaces the question of how to operationalise this in models.

See our responses below.

a. extensions of existing models The suggestion here is to adjust human impact on the environment using empirical data on the correlation between the religious value set and observed impact on the environment and also environmental feedback on human behaviour. Examples posited of the forward impact were the religious shaping of agro-forestry, pastoralism and land management practices; dietary preferences and population growth. The first four of these have certainly been important in the past but it might be argued that, in the future, satisfying the demand for food and living space will swamp religious traditions. Collective choices on reproduction, for example through cultural expectations for large families, clearly affect the population and thence impact through IPAT but government funded education or support for fertility control and simple per capita wealth probably has more ultimate influence on population. Also, I found the idea that climate catastrophes would be blamed on supernatural displeasure and would also promote solidarity to be in the first instance unlikely and in the second optimistic. US evangelists tried to blame Hurricane Katrina on homosexuality without notably convincing many and at the same time Katrina apparently led to societal breakdown rather than solidarity. The main problem with this approach is that for the largest factors shaping human impact on the earth, religious values are mediated through political and economic processes.

Again thanks for pointing out these examples, all suggesting that a careful and diverse approach is needed to tackle the question of whether and how religion influences environmentally relevant decisions, viewpoints and actions. As mentioned above we will consider this in the revised manuscript – while emphasising that such multiple perspective is needed in follow-up research.

Unfortunately, when we come to pathway b, there are no concrete examples of the kind of model the authors have in mind, other than a suggestion that attributes of agents in agent based models should include religious value sets that affect land use. As a result, it is difficult to see precisely what is envisaged here.

Actually there are only very few examples (if at all) that applied such models and an agenda/models will still have to be developed. Nonetheless, we will try to be more specific in describing the possible type of models, their parameters etc., and we will do that also in relation to the comment of referee 2 who asked for a better embedding of

our envisaged models in the wider landscape of socio-cultural approaches and models.

Pathway c is illustrated rather by sets of research questions that bring religion and ethics into questions of what future scenarios we might as a species or as separate populations find acceptable. This approach might fit quiet well with the development of the SSP framework for IPCC IAMs.

We will consider this suggestion in our revision.

Summary This paper casts a refreshingly unconventional eye on modelling earth system dynamics by including socio-cultural factors as embodied in religious beliefs and practices. It advances a series of assertions about the influence of religious value sets on human impact on the planet, many of which can be challenged, or at least to which significant exceptions can be found. I believe its main weaknesses are threefold. First, the religious value sets are not as clear cut as the authors suggest and a more multi factorial approach may be needed. Second, when it comes to model pathway b, the construction of new models, the paper needs to suggest some concrete examples of the way the authors think this would work. There is a disconnect between some high level statements about how land use is land use shaped by religious practice, which may have been relevant once but are probably increasingly irrelevant as we head towards feeding and housing 10-11Bn humans. More specificity here would allow readers to grapple better with the paper. Third, one can make a strong argument that the principal way that religious value sets are implemented is via the medium of governments as they respond to their polities. The paper makes this point well early on but when it comes to describing new modelling paths, this seems to be lost. In other words, a more general formulation is required of how institutions and norms affect collective behaviour. These institutions and norms include religious (sacred and secular) value sets and thereby govern impact on the biosphere and collective reaction to biospheric changes. Trying to shortcut this level of understanding as suggested in modelling path a seems to invite false correlations and parameterisations.
Thanks for summarising this; see above text for our intended responses to these issues.

Finally, I wonder if the authors want to comment on how religious value sets affect the way society reacts to other factors that operate outside its conscious understanding. I'm thinking of factors such as those described by Tainter in his many treatises on the collapse of Complex Societies (eg. Tainter, 2006 and refs therein) once the returns to complexity start to diminish. Past societies found various explanations for what must have seemed like inexorable negative forces out of their control. Would current religious value sets help or hinder our early appreciation of the fact we are on an unsustainable path? Tainter, J. A. (2006) Social Complexity and Sustainability. Ecological Complexity. 3. 91-103

This is a very good suggestion on which we will reflect, also considering further macro-historic research such as by Karl Butzer.

---

## Author Response (AR1)

**Response to Anonymous Referee #1**

*This is an interesting and important area that needs to be included in Earth System modelling. The authors have given a good overview of religions and how the belief systems can influence the way humans are using and impacting the earth system, and the biosphere in particular.*

We are thankful for the referee's overall positive assessment of our paper.

*I agree with the three approaches suggested to further this area, but I think the last one – c) distillation of deeper humanitarian questions from religious viewpoints that can inform and guide model applications and scenario building – needs to be done at the start or in conjunction with the others.*

Thank you for pointing to this paper-structural issue. We agree that point c) is not at the same level than the two other points, as it does not provide a model avenue as such. We therefore refrain from presenting this section as another model avenue but simply as an extra section that provides suggestions for research questions which can be addressed by the discussed model approaches. The introductory paragraph of section 4.4 has been changed accordingly.

*This relates to the main area that needs expansion: The authors refer several times to how religions are interpreted, and the extent to which this makes the inclusion of religion in modelling worthwhile. They mention it again in the last paragraph. The use of religion to achieve personal or group aims is widely acknowledged. It's not only USA evangelicalism. All religions interpret the divine messages to suit their purposes. Does this enhance or modify the effect the group is having on the environment? Or does it just make (that) religion irrelevant? The paper needs expansion of this area, with some indication of how it should be approached from a research perspective.*

We now mention the potential relevance of people drawing individual conclusions from religious ideas and doctrines (possibly for their own benefit); specifically Sect. 3 has been enhanced by a statement on this ambivalence – in line with an elaboration (requested by the Editor) on the fact that individual views of people confessing to the same religion may certainly differ and contrast with the 'official' doctrine. Also, for the sake of forestalling misunderstandings, we added a differentiation early in the paper regarding the aspects of religion that we wish to investigate: see end of Sect. 1.

*I note a number of typos, but have not recorded them. The paper needs editing.*

We carefully checked the revised version for any typos etc.

**Response to review by A.M. Grisogono**

*The paper identifies an important lacuna in the literature on earth system modelling and analysis, and proposes a brave approach for addressing it objectively, building on existing knowledge bases and techniques.*

Thank you for this overall positive assessment of our paper.

*It is necessarily a risky enterprise given the potential for strongly emotional and subjective reactions from fervent adherents of particular beliefs, whether through challenging the global impact of their values, or fuelling a 'blame game' that may further sharpen the divisive polarisation we already see in national and international politics. Nevertheless, the authors are correct in pointing out the risks of overlooking the impacts of the geographical distributions and strengths of different religious value sets on how human societies interact with environmental and climate change.*

Indeed, as you note, we tried to address the possible and diverse roles of religion for global sustainability (and models of the earth system) as objective and careful as possible, pointing to both the positive and the negative sides. We stress these even more in the revised version, also in response to more specific comments by the other Reviewer and the Editor .

*Of course, religion is not the only cultural determinant that influences the climate change relevant behaviours of social groups, and it would be good to see how to set the proposed research program into a wider socio-cultural context, which might ultimately provide a more comprehensive account of the complex trajectories we observe and wish to influence. My main concern is how these 'sacred cows' can be treated in a balanced and rational way with scientific integrity, without provoking a destructive backlash.*

In some places of the paper we strengthened our (already existing) statements that the suggested modelling of religion as a marker can be seen as a part of the wider socio-cultural modelling approaches for Earth system modelling, especially at the end of Sect. 4.3 where we relate to already tested (e.g. socio-hydrological) modelling approaches. We also slightly enhanced the final paragraph of the Conclusion by your suggestion to ensure that scientific inquiry of religious attitudes and their environmental relevance be neutral and free of subjective statements (without potentially "provoking a destructive backlash" in the real world).

*The text would benefit from some minor editing, as noted by the other referee.*

We carefully edited the revised version.

**Final response to Editor John Finnigan**

*This paper argues persuasively for the inclusion of important socio-cultural factors in earth system models, particularly in the IAMs used to design and test human policy responses to climate change mitigation and adaptation. It focusses on religion(s) as the primary shapers of human decision making at population/voting bloc level and teases apart the attitudes of different religions and sub sects to environmental protection and the use of earth's resources. The aims of the paper are threefold: 1. To explain how religion(s) can serve as markers for modern culture so that their inclusion can add a deeper human dimension to earth system modelling. 2. To propose environmental 'value sets' to capture aspects of religion in modelling the whole earth system (understood here as the intersection of the social and biophysical worlds). 3. To provide some initial research primers for quantifying the required steps. I think the paper succeeds quite well in approaching all three aims but leaves open a serious gap between the influence of religion on human attitudes and the way these attitudes come to be expressed in decision and policy making. I'll expand on this view below.*

Thanks for the elaborate and thoughtful comments; we addressed them as follows.

*1. religion(s) as markers for modern culture*

*It makes sense to adopt the 3rd widest definition of religion, that is, as a belief in the sacred but I wonder if the definition of 'sacred' also needs to be wider? Communism was effectively a set of beliefs about the optimal way of organising social interaction and 'the means of production', which were received and largely unquestioned by generations in the eastern bloc post 1917. Capitalism and the primacy of the individual has held the same position in the USA through most of the 20th century and today. Economic and cultural nationalism (autarky) had a powerful and ultimately disastrous airing between the 2 world wars and seems to have reappeared on the world stage today. More recently, a sub sect of capitalism, neo-liberal economics, has held sway in most Anglophone polities for the last 45 years. A feature of both neo-liberalism and communism has been the unquestioned adherence of its proponents to ideas for which there is no empirical evidence or which are contradicted by historical and current data (eg. trickle down economics). More important for the arguments in this paper is that these 'secular religions' directly shape policy whereas more traditional religious beliefs have to be filtered through a further step of electing representatives who profess those beliefs, at least in democracies. Communism, at least in its soviet interpretation of Engel's dialectic materialism, had disastrous environmental impacts. The untrammelled application of neo liberal economics today seems to be producing the same results but based on a totally different philosophy.*

Thank you for the broad cultural comparison and the bringing into play of 'secular' religions. We have revised the manuscript, especially Sect. 3, in various aspects including a somewhat more precise definition of how we use the term 'religion' and a more detailed consideration of the interplay of religion and politics.

*2. environmental value sets*

*this is a useful survey of the attitudes of different major sacred (as opposed to secular) religions to the environment and to humanity's use of natural resources. However, I wonder if the authors' assessment of the role of religion is not more positive than is justified. For example, the papal encyclical 'Laudato Si' certainly said all the right things but it came from a Pope whose liberal views are strongly opposed by many powerful conservatives in the church hierarchy. The liberal-conservative battle in the catholic church is playing out on several fronts with cover ups of child sexual abuse and the role of women being major battle grounds but conservation and climate change*

*is also far from accepted by conservative Catholics. The 'number two' in the Vatican hierarchy, Cardinal George Pell is a prominent climate denier and many other examples of powerful conservative figures who oppose Laudato Si can be found. Hence, I do not wholly agree that the catholic church forms one pole of the Christian spectrum with regard to environmental and climate attitudes, evangelical Protestantism being the other. Rather it seems to me that conservatives in both catholic and protestant churches might have more in common with each other than with liberals in their own churches. Similarly, conservative Islam has many attitudes in common with catholic and protestant conservatives in areas like birth control and creationism so it is questionable quite how well the major religious divisions used in the paper map onto value sets.*

We agree that the juxtaposition of Roman Catholic and Evangelical positions was too simple. We now tried to provide a more balanced view (while still arguing that specific environmental value sets associated with specific religions are suited for model exercises) – see Sect. 3, where we also briefly address the recent critique of *Laudato Si'* from Catholic opponents as well as the fact that individual positions of people confessing to the same religion can certainly differ. In general, we now better highlight the intra- and inter-religious ambivalences in several places of the revised manuscript.

*There are a few other statement in this section of the ms that could be challenged. For example, that religion fosters social connectivity and so has the potential to create empathy for people in distant places and future generations. Religions are also powerful ways to define and exclude 'the other' so that adherents don't have to feel empathy. There are of course ample examples from history but the demonization of the poor by US evangelical Prosperity Theology is a good current example of how a religion can be used to avoid empathy with the disadvantaged. In summary, I found this section a good first attempt at identifying religions and value sets but the objections I raise above suggest that a subtler multi-factorial classification might be needed to operationalise this concept in any model.*

As stated in the previous response, we have revised some parts of the paper, to provide a more balanced view and suggest the need for multifactorial approaches (see especially Sect. 3 and 4.3). We thereby also strengthen the point we made, that systematic databases on which such modelling can be based are still a desiderate (though stylised model experiments are still possible even in the absence of comprehensive empirical data).

*3. Modelling avenues*

*Three possible pathways are considered: a) extensions of existing models; b) design of new model types; c) Distillation of deeper humanitarian questions from religious viewpoints that can inform and guide modelling. The authors suggest that religion can be considered a social technology in an IPAT formulation of environmental impact of society. This is unarguable but simply displaces the question of how to operationalise this in models.*

*a. extensions of existing models*

*The suggestion here is to adjust human impact on the environment using empirical data on the correlation between the religious value set and observed impact on the environment and also environmental feedback on human behaviour. Examples posited of the forward impact were the religious shaping of agro-forestry, pastoralism and land management practices; dietary preferences and population growth. The first four of these have certainly been important in the past but it might be argued that, in the future, satisfying the demand for food and living space will swamp religious traditions. Collective choices on reproduction, for example through cultural expectations for large families, clearly affect the population and thence impact through IPAT but government funded*

*education or support for fertility control and simple per capita wealth probably has more ultimate influence on population. Also, I found the idea that climate catastrophes would be blamed on supernatural displeasure and would also promote solidarity to be in the first instance unlikely and in the second optimistic. US evangelists tried to blame Hurricane Katrina on homosexuality without notably convincing many and at the same time Katrina apparently led to societal breakdown rather than solidarity. The main problem with this approach is that for the largest factors shaping human impact on the earth, religious values are mediated through political and economic processes.*

We revised Sect. 4.2 accordingly, as we agree that some of our former suggestions were superfluous; we also added a paragraph on the need to investigate whether the possible influence of religion on these processes is actually diminishing.

*Unfortunately, when we come to pathway b, there are no concrete examples of the kind of model the authors have in mind, other than a suggestion that attributes of agents in agent based models should include religious value sets that affect land use. As a result, it is difficult to see precisely what is envisaged here.*

Actually there are only very few examples that applied such models and an agenda/models will still have to be developed. But we now tried to be more specific in describing possible model types, along with addressing the request by referee 2 to better embed these in the wider landscape of socio-cultural approaches and models. See the revision of Sect. 4.3.

*Pathway c is illustrated rather by sets of research questions that bring religion and ethics into questions of what future scenarios we might as a species or as separate populations find acceptable. This approach might fit quiet well with the development of the SSP framework for IPCC IAMs.*

We reformulate: the former "pathway c" is no longer treated as a modelling approach but as a collection of research question to be addressed by models of type a) or b). We also add the remark that this can be complementary to the development of SSP storylines (Sect. 4.4).

*Summary*

*This paper casts a refreshingly unconventional eye on modelling earth system dynamics by including socio-cultural factors as embodied in religious beliefs and practices. It advances a series of assertions about the influence of religious value sets on human impact on the planet, many of which can be challenged, or at least to which significant exceptions can be found. I believe its main weaknesses are threefold. First, the religious value sets are not as clear cut as the authors suggest and a more multi factorial approach may be needed. Second, when it comes to model pathway b, the construction of new models, the paper needs to suggest some concrete examples of the way the authors think this would work. There is a disconnect between some high level statements about how land use is land use shaped by religious practice, which may have been relevant once but are probably increasingly irrelevant as we head towards feeding and housing 10-11Bn humans. More specificity here would allow readers to grapple better with the paper. Third, one can make a strong argument that the principal way that religious value sets are implemented is via the medium of governments as they respond to their polities. The paper makes this point well early on but when it comes to describing new modelling paths, this seems to be lost. In other words, a more general formulation is required of how institutions and norms affect collective behaviour. These institutions and norms include religious*

*(sacred and secular) value sets and thereby govern impact on the biosphere and collective reaction to biospheric changes. Trying to shortcut this level of understanding as suggested in modelling path a seems to invite false correlations and parameterisations.*

The points raised in this summary are now individually addressed as described above. In particular, we agree that the "shortcut" issue may arise and should be prevented by constructing a well-defined database which disentangles religion-motivated paths of action from other sources. Yet it is also common practice in (natural science-based) Earth system modelling to allow for emulator functions on any level of detail where there is a dearth of data or a lack of deeper understanding – which could also be useful here, wilfully accepting the "shortcut" as a necessary means of simplification.

*Finally, I wonder if the authors want to comment on how religious value sets affect the way society reacts to other factors that operate outside its conscious understanding. I'm thinking of factors such as those described by Tainter in his many treatises on the collapse of Complex Societies (eg. Tainter, 2006 and refs therein) once the returns to complexity start to diminish. Past societies found various explanations for what must have seemed like inexorable negative forces out of their control. Would current religious value sets help or hinder our early appreciation of the fact we are on an unsustainable path? Tainter, J. A. (2006) Social Complexity and Sustainability. Ecological Complexity. 3. 91-103*

We think it is adequate to point to this reference in one of the research questions listed in Sect. 4.4.